# Dedicated transcriptomics combined with power analysis lead to functional understanding of genes with weak phenotypic changes in knockout lines

**Chen Xie** [ORCID]*, **Cemalettin Bekpen**[¤a], **Sven Künzel**, **Maryam Keshavarz** [ORCID], **Rebecca Krebs-Wheaton** [ORCID], **Neva Skrabar**[¤b], **Kristian K. Ullrich** [ORCID], **Wenyu Zhang** [ORCID], **Diethard Tautz** [ORCID]

Department of Evolutionary Genetics, Max Planck Institute for Evolutionary Biology, Plön, Germany

¤a  Current Address: Department of Molecular Biology and Genetics, Bahcesehir University, Istanbul, Turkey
¤b  Current Address: International Centre for Genetic Engineering and Biotechnology, Trieste, Italy
*  xie@evolbio.mpg.de

**Data Availability Statement:** All RNA-Seq files are available from the European Nucleotide Archive database (accession number PRJEB28348).

## Abstract

Systematic knockout studies in mice have shown that a large fraction of the gene replacements show no lethal or other overt phenotypes. This has led to the development of more refined analysis schemes, including physiological, behavioral, developmental and cytological tests. However, transcriptomic analyses have not yet been systematically evaluated for non-lethal knockouts. We conducted a power analysis to determine the experimental conditions under which even small changes in transcript levels can be reliably traced. We have applied this to two gene disruption lines of genes for which no function was known so far. Dedicated phenotyping tests informed by the tissues and stages of highest expression of the two genes show small effects on the tested phenotypes. For the transcriptome analysis of these stages and tissues, we used a prior power analysis to determine the number of biological replicates and the sequencing depth. We find that under these conditions, the knockouts have a significant impact on the transcriptional networks, with thousands of genes showing small transcriptional changes. GO analysis suggests that *A930004D18Rik* is involved in developmental processes through contributing to protein complexes, and *A830005F24Rik* in extracellular matrix functions. Subsampling analysis of the data reveals that the increase in the number of biological replicates was more important that increasing the sequencing depth to arrive at these results. Hence, our proof-of-principle experiment suggests that transcriptomic analysis is indeed an option to study gene functions of genes with weak or no traceable phenotypic effects and it provides the boundary conditions under which this is possible.

## Author summary

Knockout mice benefit the understanding of gene functions in mammals. However, it has proven difficult for many genes to identify clear phenotypes, related due to lack of

**Funding:** This work was supported by a European Research Council advanced grant (https://erc.europa.eu/funding/advanced-grants) to DT (NewGenes - 322564). The funders had no role in study design, data collection and analysis, decision to publish, or preparation of the manuscript.

**Competing interests:** The authors have declared that no competing interests exist.

sufficient assays. As Lewis Wolpert put it in a famous quote "But did you take them to the opera?", thus metaphorically alluding to the need to extend phenotyping efforts. This insight led to the establishment of phenotyping pipelines that are nowadays routinely used to characterize knock-out lines. However, transcriptomic approaches based on RNA-Seq have been much less explored for such deep-level studies. We conducted here both, a theoretical power analysis and practical RNA-Seq experiments on two knockout lines with small phenotypic effects to investigate the parameters including sample size, sequencing depth, fold change, and dispersion. Our dedicated RNA-Seq studies discovered thousands of genes with small transcriptional changes and enriched in specific functions in both knockout lines. We find that it is more important to increase the number of samples than to increase the sequencing depth. Our work shows that a deep RNA-Seq study on knockouts is powerful for understanding gene functions in cases of weak phenotypic effects, and provides a guideline for the experimental design of such studies.

## Introduction

When the generation of specific gene knockouts became possible in mice, it was soon noticed that not all knockouts had obvious phenotypic effects. This included also cases where highly conserved genes were deleted, which would have been thought to play a major role in cell physiology or development. These findings have led to the notion that there must be some developmental buffering or redundancy in biological systems that can compensate for the loss of single genes [1]. Initially, it was thought that this redundancy could be due to gene duplications, but this expectation has only partially been verified [2,3]. When the initial observations on the lack of phenotypes of knockout mutants were made, Lewis Wolpert used to ask at conferences: "But did you take it to the opera?" implying that looking only for obvious phenotypes is not sufficient to understand the functions of genes. After all, functions are optimized in the course of evolution and even small selective advantages can lead to functions, especially under different environmental conditions. This discussion, which was much influenced by Wolpert´s quote, led to the conclusion that the equivalent of "operas" had to be built to better understand the function of genes. Of course, not an opera was required, but a much broader scale of phenotyping efforts. In the case of the mouse, this challenge has been taken on by the International Mouse Phenotyping Consortium (IMPC), which aims to characterize up to 5,000 knockout lines based on a standardized high-throughput pipeline [4,5]. However, the systematic use of transcriptome evidence is not envisaged by this consortium so far. Systematic transcriptomic evidence is currently only collected by the "Deciphering the Mechanisms of Developmental Disorders (DMDD)" program, but with a focus on embryonic lethal mutant lines [6], i.e. will not provide further insights into the cases where a knockout shows no overt phenotype.

We argue here that deep transcriptome comparisons may yield results for knockout lines where no or only small phenotypic difference between knockouts and wildtypes were detected by the tests employed in the established phenotyping pipelines [5]. To generate a proof-of-principle, we have studied two IMPC project lines. Given our interests in *de novo* gene evolution [7], we have chosen knockouts of two mouse young genes, *i.e.*, genes that have likely emerged out of ancestral non-coding regions, which can a priori be expected to show only weak phenotypes [8]. Both knockout lines are indeed homozygous viable with no overt phenotype, but the IMPC phenotyping pipeline has generated initial evidence for weak phenotypes in one of them. Apart of doing further phenotyping tests, we decided to do a systematic

transcriptome analysis for these strains. Since the knockout lines show only weak phenotypes, we did not expect that transcriptome data would show dramatic differences in terms of large numbers of differentially expressed genes with fold changes larger than 1.5 or 2. Hence, it was crucial to develop a dedicated power analysis for transcriptome data. Statistics for RNA-Seq studies have been well established [9,10], and power analysis has also been discussed and studied theoretically [11–14]. However, deep practical studies about the power analysis with real data are still lacking. We show here that more samples and deeper sequencing is required than usually applied in transcriptome studies so far, whereby the increase in the number of samples is more important than the increase in sequencing depth. When conducting such deeper analysis, we find that both loci significantly impact many other genes in the transcriptomic networks of the respective tissues. We suggest that this deeper transcriptomic approach–the equivalent of taking the "mice to the transcriptomic opera"–has the potential to contribute to a better understanding of gene function and the role of interacting networks in generating phenotypes.

## Results

### Genes for functional analyses

The two loci chosen for this study (*A930004D18Rik* and *A830005F24Rik*) were derived from a list of newly evolved genes in the house mouse that we have established in an effort to study *de novo* gene evolution [15]. This list was generated based on searches for translated ORFs that are only found in the mouse reference genome, but not in rat or other mammals. We validated their annotated gene structures using Illumina RNA-Seq and PacBio Iso-Seq data, we obtained their transcriptional expression profile using the strand-specific ENCODE RNA-Seq data in 35 tissues and we confirmed the translation of their predicted open reading frames (ORFs) using the strand-specific ribosome profiling (Ribo-Seq) data.

 *A930004D18Rik* is annotated with two exons in the main gene annotation databases, including Ensembl (versions from 80 to 98), NCBI, and UCSC (Fig 1A). However, our reanalysis of the available transcriptomic data did not confirm the annotated first exon. There is not even a single read from the ENCODE RNA-Seq data in 35 tissues, our strand-specific Illumina RNA-Seq data presented below, or our strand-specific PacBio Iso-Seq data from mouse brains (see Methods) that would support the existence of the first exon. Tracing the origin of this annotation in the UCSC genome browser revealed that it is only supported by a single spliced EST (BB642260) and a single cDNA (AK044329) from a retina library, *i.e.*, this first exon may be very tissue specific or was an artefact in the respective library. In contrast, both PacBio Iso-Seq and Illumina RNA-Seq data confirmed the second exon (Fig 1A). PacBio Iso-Seq result shows a single-exon transcript (Isoform1). Illumina RNA-Seq supports two additional double-exon transcripts with different first exons (Isoform2 and Isoform3) (Fig 1A). Considering the complicated transcription of *A930004D18Rik*, we used its annotated second exon to estimate its expression profile among tissues. It shows a relatively high expression (up to FPKM 3.9) in multiple tissues, with the highest in brain parts at different stages, as well as in embryonic limbs (Fig 1C and S1 Table). The annotated ORF (ORF2 in Fig 1A) was validated with Ribo-Seq triplet periodicity. Furthermore, we found an additional upstream ORF (ORF1 in Fig 1A) to be also supported by the Ribo-Seq triplet periodicity data, making this a potentially bi-cistronic gene [16]. Based on genome alignment data from various species, it is possible to suggest that the two ORFs have emerged at different times: 2–4 million years ago (mya) for ORF1 and 8–10 mya for ORF2 (S1 Fig). The knockout construct leads to a replacement of the entire second annotated exon including ORF1 and ORF2 by *LacZ* (Fig 1A, Methods). We guess that IMPC conducted this knockout strategy because it was assumed that *A930004D18Rik* would

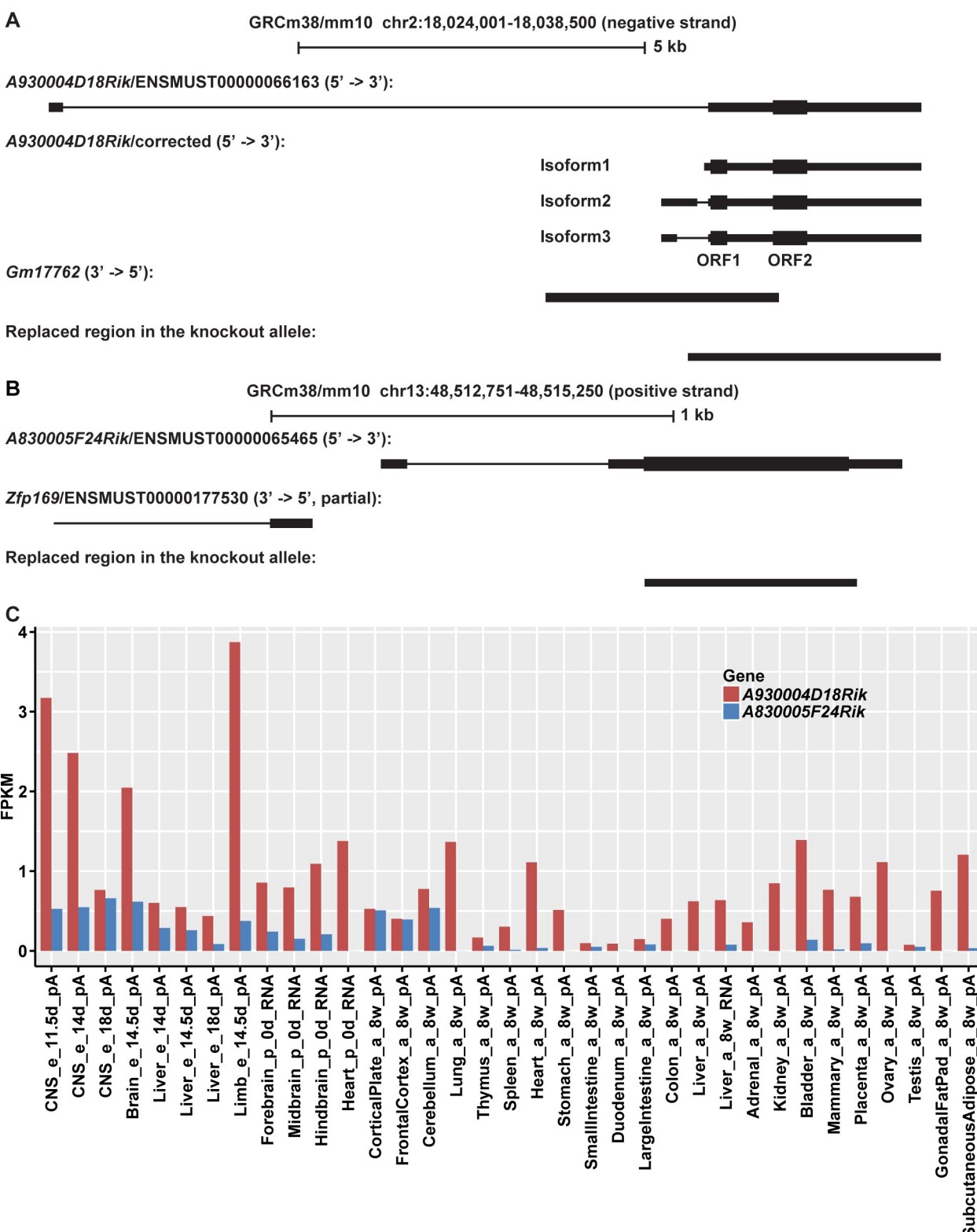

**Fig 1. Genomic contexts and expression profiles of *A930004D18Rik* and *A830005F24Rik*.** (A) Transcript structures of *A930004D18Rik*. The annotated structure of *A930004D18Rik* is shown as ENSMUST00000066163. (B) Transcript structures of *A830005F24Rik*. The exact positions of the elements shown in (A) and (B) are provided in BED format in S1 Table. (C) Expression profiles of *A930004D18Rik* and *A830005F24Rik* derived from the ENCODE RNA-Seq data in 35 tissues. The sample information and exact FPKM values are provided in S1 Table.

be a long non-coding RNA, but the targeting strategy for this specific line could not be validated in retrospect by IMPC or the European Mouse Mutant Archive (EMMA). According to the information from gene annotation databases and the evidence from Illumina RNA-Seq and PacBio Iso-Seq data, a long non-coding RNA (*Gm17762*) on the reverse strand overlaps with the replaced region (Fig 1A).

The transcript of *A830005F24Rik* includes two exons and one ORF (Fig 1B). Our reanalysis of all available transcriptome data (see above) confirms this annotation. It seems possible that it has emerged through bidirectional transcription, a mechanism that has been postulated to be involved in *de novo* transcript formation [17,18]. The neighboring gene *Zfp169* is transcribed in the opposite direction and the transcription start sites are only ~100 bp apart of each other (Fig 1B). It shows on average a lower expression (up to FPKM 0.7), mostly in brain parts at different stages (Fig 1C and S1 Table). The annotated ORF was confirmed with Ribo-Seq triplet periodicity. This ORF is predicted to have emerged 6–8 mya based on the genome alignments (S1 Fig). The knockout construct leads to a replacement of most of this ORF by *LacZ* (Fig 1B, Methods).

### *A930004D18Rik* knockout phenotyping

The relatively high expression of *A930004D18Rik* in the CNS indicated that it could have an effect on the behavior of the mice. Hence, we performed three standardized behavioral tests: elevated plus maze, open field, and novel object (each with several parameters, Table 1). We found a significant difference for the open field test with respect to total distance moved (nested ranks test, corrected P-Value (BH) = 0.018; Tables 1 and S2).

Given that *A930004D18Rik* is also highly expressed in limbs, we asked whether there would also be differences in limb morphology. To assess a possible limb phenotype, we scanned the

**Table 1.  Phenotyping results for *A930004D18Rik*.**

| Test | Parameter | N[a] | KO[b] | WT[b] | P-Value[c] | Corrected P-Value[d] |
|---|---|---|---|---|---|---|
| Elevated plus maze | center time (%) | 40 | 11.9 | 10.8 | 0.19 | 0.30 |
|  | dark time (%) | 40 | 54.1 | 56.7 | 0.20 | 0.30 |
|  | light time (%) | 40 | 31.0 | 28.5 | 0.15 | 0.30 |
| Open field | wall time (%) | 40 | 51.4 | 44.7 | 0.24 | 0.30 |
|  | total distance (m) | 40 | 42.1 | 48.0 | 0.0023 | 0.018 |
| Novel object | first contact time (s) | 40 | 2.5 | 5.0 | 0.26 | 0.30 |
|  | object visits (N) | 40 | 4.0 | 3.0 | 0.14 | 0.30 |
|  | total distance (m) | 40 | 28.2 | 30.1 | 0.35 | 0.35 |
| Limb elements (length in mm) | humerus | 40 | 11.96 | 11.96 | 0.93 | 0.93 |
|  | ulna | 40 | 13.86 | 13.83 | 0.37 | 0.44 |
|  | metacarpal | 40 | 3.20 | 3.22 | 0.043 | 0.13 |
|  | femur | 40 | 15.34 | 15.44 | 0.21 | 0.32 |
|  | tibia | 40 | 17.37 | 17.21 | 0.072 | 0.14 |
|  | metatarsal | 40 | 7.43 | 7.29 | 0.020 | 0.12 |
|  | PC3 (8.8% variance) | 40 | 0.688 | -0.644 | $5.4 \times 10^{-9}$ | $3.2 \times 10^{-8}$ |

[a]N = total number of individuals used, equally divided between knockouts and wildtypes.

[b]Medians across all individuals.

[c]P-Values for the behavior phenotypes were calculated using nested ranks test representing a non-parametric linear mixed model. For the limb length measurements, we used two-tailed Wilcoxon rank sum tests.

[d]Benjamini-Hochberg correction applied to eight tests for the behavior analysis, to six tests for the bone length and to six PC axis for the PCA.

skeletons of the respective knockout (KO) and wildtype (WT) mice and analyzed their bone lengths, following the procedures described in [19]. We found that the knockout mice tended to have longer metatarsals (two-tailed Wilcoxon rank sum test, corrected P-Value (BH) = 0.12), shorter metacarpals (two-tailed Wilcoxon rank sum test, corrected P-Value (BH) = 0.13), and longer tibias (two-tailed Wilcoxon rank sum test, corrected P-Value (BH) = 0.14; Tables 1 and S2). Given that three out of six tested bones showed these tendencies, but without reaching significance, we conducted a principle component analysis (PCA). We found that PC3, which explains 8.8% variance, separated the genotypes well (two-tailed Wilcoxon rank sum test, corrected P-Value (BH) = $3.2 \times 10^{-8}$; Table 1 and S2 Fig).

### *A830005F24Rik* knockout effect on mouse behavior

The *A830005F24Rik* knockouts were preliminarily phenotyped by the International Mouse Phenotyping Consortium phenotyping pipeline (IMPC). We reanalyzed all phenotypic data, and found 11 parameters that showed tendencies (S3 Table and S1 File). Based on this, we did eight preliminary tests on a restricted number of animals in order to confirm possible tendencies (S3 Table and S1 File). For one test (elevated plus maze), we expanded the number of test animals and measured all three parameters for this test (Table 2). We found that the time spent in the center was shorter, both in the expanded test, as well as when all individuals were considered (nested ranks test, corrected P-Value (BH) = 0.10 or 0.086; Tables 2 and S2). The center time difference indicates a decision-making related phenotype [20–22].

### Power analysis for transcriptomic data

Gene effects cannot only be traced via their phenotypes, but also via their effects they leave on the transcriptomic network. However, this option has not been much explored so far for knockouts with weak phenotypic effects. We propose here to expand the repertoire of systematic gene effect studies also to transcriptomic networks. To investigate the experimental boundary conditions for this, we have first done a power analysis based on exploring the parameter space using RNASeqSampleSize (1.6.0) [11]. Several conditions have to be considered for such a power analysis. When using RNA-Seq read count (fragment count for paired-end sequencing) data, we assume (1) that read counts follow a negative binomial distribution; (2) that all samples are sequenced at the same depth; (3) that the significance cutoff after Bonferroni correction is set to P-Value ≤ 0.05 (*i.e.*, when a total of 15,000 genes are tested, the significance cutoff before adjustment is P-Value ≤ $3.3 \times 10^{-6}$). The power to detect a differentially expressed gene (DEG) can then be estimated by the (1) sample size, (2) fold change between knockouts and wildtypes, (3) average read count, and (4) dispersion, which is the measurement of biological and technical variance considering the effect of average read

**Table 2. Phenotyping results for *A830005F24Rik*.**

| Test | Parameter | N[a] | KO[b] | WT[b] | P-Value[c] | Corrected P-Value[d] | P-Value (expanded only, N = 24) | Corrected P-Value[d] (expanded only, N = 24) |
|---|---|---|---|---|---|---|---|---|
| Elevated plus maze | center time (%) | 36 | 10.8 | 14.8 | 0.029 | 0.086 | 0.035 | 0.10 |
| | dark time (%) | 36 | 63.2 | 58.3 | 0.072 | 0.11 | 0.18 | 0.27 |
| | light time (%) | 36 | 21.7 | 20.5 | 0.45 | 0.45 | 0.74 | 0.74 |

[a]N = total number of individuals used, equally divided between knockouts and wildtypes.

[b]Medians across all individuals.

[c]P-Values were calculated using nested ranks test representing a non-parametric linear mixed model.

[d]Benjamini-Hochberg correction applied to the three parameters of the elevated plus maze test.

count. The dependencies of these four factors are graphically summarized in Fig 2A. Since we were particularly interested in studying weak effects, we focus on the parameter space that covers fold changes up to 1.5 (X-axis in Fig 2A; note that fold-changes below 1 would yield symmetrical results). This is a parameter space that is rarely explored, since standard transcriptomic procedures tend to use the arbitrary cutoff of 2-fold change. But increasing the number of biological replicates (sample size), the depth of sequencing (read count), and controlling the experimental and biological variance (dispersion), one can indeed obtain enough power for DEG with low fold changes (see colored dots in Fig 2A). This shows that for genes with low fold change (from 1.05 to 1.25) that larger number of biological replicates, deeper depth of sequencing, and smaller variances increase the power much more than for genes with higher fold change (Fig 2A). Note that the Bonferroni correction we used here is usually considered to be too conservative for transcriptome data, and FDR correction should be applied. However, FDR correction is based on the P-Value distribution of all the tested genes, which is not suitable for this particular analysis. We applied FDR correction (BH) for all other analyses below. Even though the multiple testing control is too strict here, the results show that genes with very low fold changes can still be detected when the condition of other parameters (such as sample size, read count, and dispersion) are adjusted.

Based on this analysis, we used at least 10 biological replicates of knockouts and wildtypes each, performed deep sequencing and minimized variance by using standardized rearing conditions for the mice, as well as standardized and parallel RNA preparation and sequencing procedures. This allowed indeed to identify the perturbations of the transcriptomic network in the knockout mice of the two tested loci (Fig 3 and S6 Table, see further details below).

In order to confirm the effects of sample size and sequencing depth on discovering DEGs with low fold change, we performed two sets of random subsampling studies on our RNA-Seq dataset "*A930004D18Rik* embryo female" for which we collected the largest sample size (Fig 3A). The first set of subsampling was on sample size. We subsampled N knockout and N wild-type samples (N from 3 to 10) from the total set of 12/14 KO/WT samples and identified DEGs (DESeq2, adjusted P-Value ≤ 0.05) each time, and repeated this 1,000 times for each N value. The number of detected DEGs increased continuously with the number of samples, suggesting that a saturation was not yet reached (Fig 2B). The second subsampling was on sequencing depth. We subsampled fragments (a fragment is a pair of reads) from total fragments of all genes in each sample with a ratio from 0.1 to 0.9 and identified DEGs (DESeq2, adjusted P-Value ≤ 0.05) each time, and repeated this 1,000 times for each ratio value. For this subsampling we found also a continuous increase in DEGs detected, but more akin to a saturation curve (Fig 2C). This suggests that raising the number of samples could be more efficient than raising sequencing depth for detecting the maximum number of DEGs.

To further understand the effects of sample size and sequencing depth on downstream functional analysis, we performed Gene Ontology (GO) enrichment analysis with GOseq (1.38.0) [23] using the outputs of the differential expression analyses described in the paragraph above. The proportion of overlapping GO terms between the significantly enriched terms (BH corrected P-Value ≤ 0.05) in the subsampling dataset and those in the full dataset was used as the measurement of the robustness, which is defined as the smaller ratio between the number of overlapping terms divided by the number of terms enriched in the subsampling dataset and the number of overlapping terms divided by the number of terms enriched in the full dataset. As shown in Fig 2D and 2E, the more samples or the deeper sequencing, the larger the proportion of overlapping GO terms that was discovered. In addition, the proportions of overlapping GO terms of the sample size subsampling datasets are smaller than those of the sequencing depth subsampling datasets, which indicates that raising the number of samples could be more efficient than raising sequencing depth for accurately detecting functional

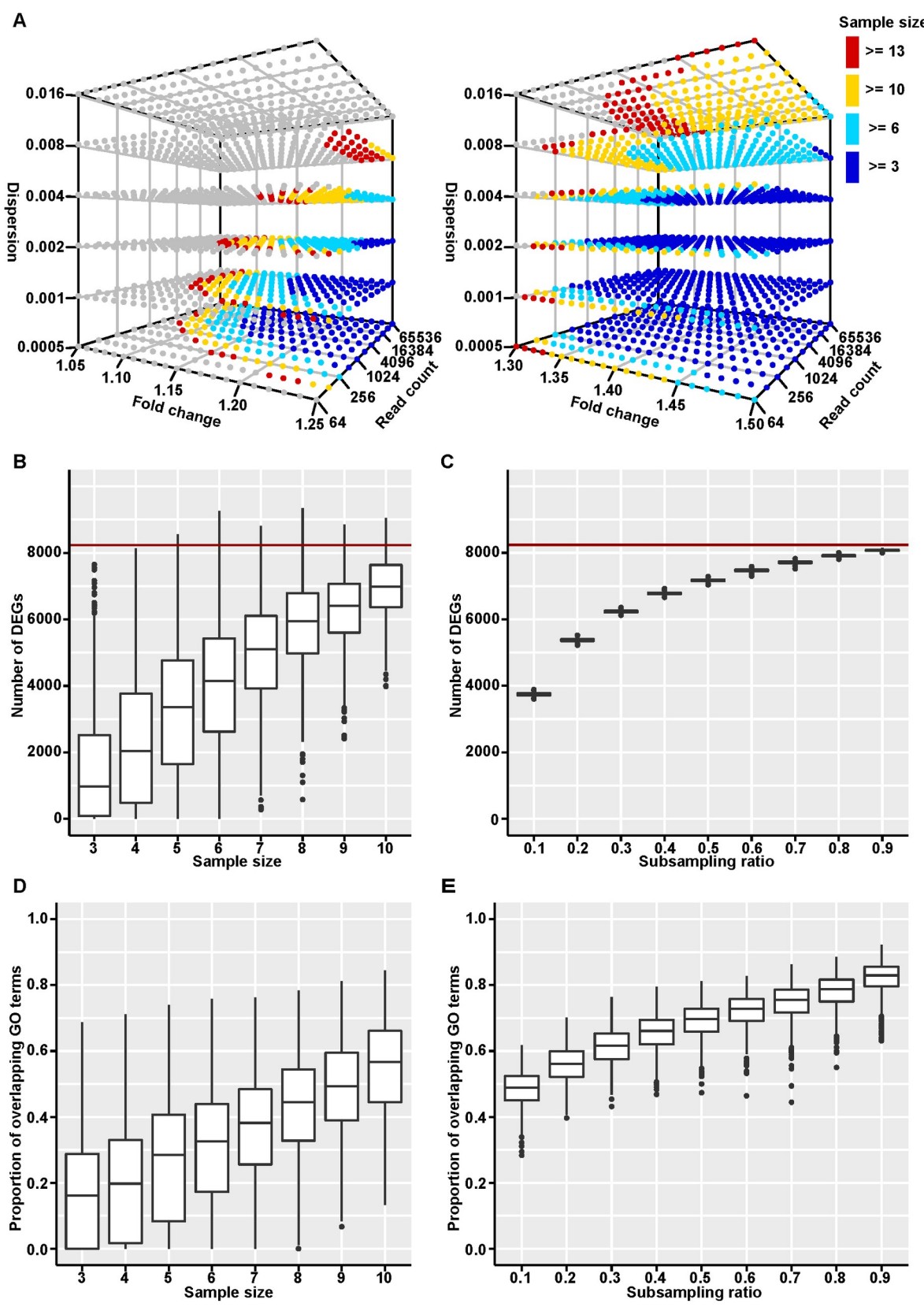

**Fig 2. Transcriptomic power analysis and subsampling analyses.** (A) shows the calculated power for each combination of sample size, fold change, read count, and dispersion. The three axes represent fold change, read count, and dispersion separately. The grey dots represent power lower than 0.8, and the colored dots represent power greater than or equal to 0.8 under different sample sizes. The left panel shows fold change from 1.05 to 1.25, and the right shows fold change from 1.3 to 1.5. The corresponding actual values are in S4 Table. The numbers of DEGs detected from the subsampling datasets on sample size (B) and sequencing depth (C) of our RNA-Seq dataset "*A930004D18Rik* embryo female" are shown in box plots (data in S5 Table). The red lines indicate the number of DEGs found for the full dataset. The proportions of overlapping GO terms detected from the subsampling datasets on sample size (D) and sequencing depth (E) of our RNA-Seq dataset "*A930004D18Rik* embryo female" are shown in box plots (data in S2–S4 Files). The definition of the proportion of overlapping GO terms is stated in the main text.

terms. Overall, the results of enrichment analyses on subsampling datasets are consistent with the subsampling results focusing on the numbers of DEGs presented above. This is an expected outcome, since the DEGs and their statistics including P-Values, fold changes, and read counts, are the basis of downstream analysis.

### *A930004D18Rik* knockout effect on the transcriptome

*A930004D18Rik* is broadly expressed across developmental stages and tissues (Fig 1C). High expression in brain tissues is seen in embryos and pups and the limbs in embryos.

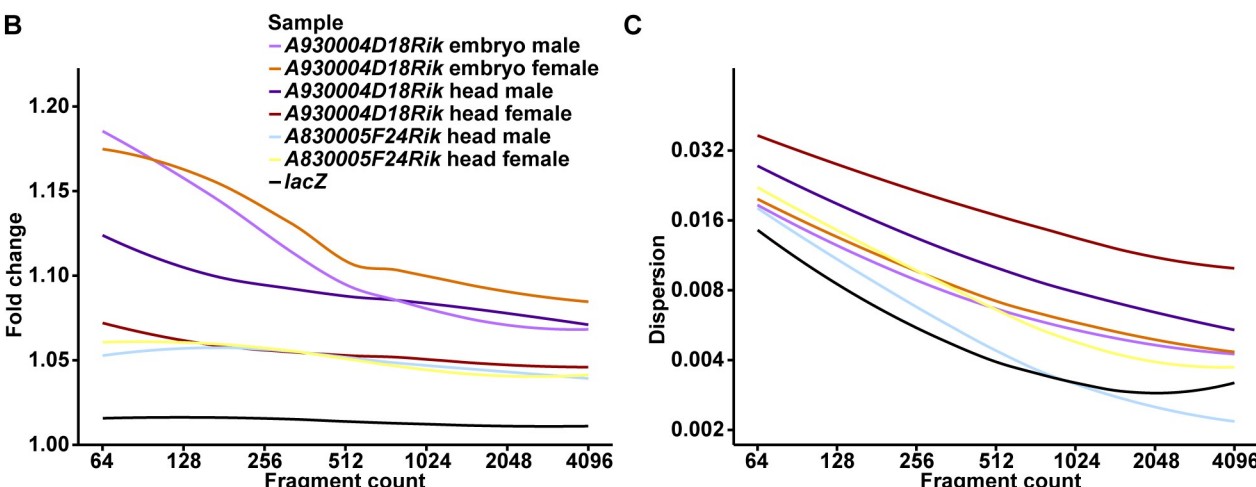

**A**

| Sample | Sample size (KO/WT) | DESeq2 | edgeR | limma | Common | DESeq2 with fold change cutoff | | | | | |
|---|---|---|---|---|---|---|---|---|---|---|---|
| | | | | | | 1.1 | 1.2 | 1.3 | 1.4 | 1.5 | 2 |
| *A930004D18Rik* embryo male | 12/10 | 6,173 | 6,104 | 6,029 | 5,803 | 4,745 | 2,049 | 893 | 384 | 146 | 1 |
| *A930004D18Rik* embryo female | 12/14 | 8,239 | 8,178 | 8,112 | 7,898 | 6,361 | 2,458 | 988 | 385 | 165 | 1 |
| *A930004D18Rik* head male (pup) | 10/10 | 4,317 | 3,719 | 3,490 | 3,337 | 3,441 | 1,086 | 233 | 35 | 4 | 1 |
| *A930004D18Rik* head female (pup) | 10/10 | 1 | 2 | 1 | 1 | 1 | 1 | 1 | 1 | 1 | 1 |
| *A830005F24Rik* head male (pup) | 10/10 | 2,733 | 2,286 | 2,278 | 2,090 | 1,421 | 92 | 7 | 0 | 0 | 0 |
| *A830005F24Rik* head female (pup) | 10/10 | 820 | 522 | 142 | 139 | 613 | 114 | 5 | 0 | 0 | 0 |
| *lacZ* | 10/10 | 0 | 0 | 0 | 0 | 0 | 0 | 0 | 0 | 0 | 0 |

**Fig 3. Summary of the differential expression analyses.** RNA-Seq datasets of *A930004D18Rik* and *A830005F24Rik* (knockouts vs. wildtypes from mouse knockout lines), and *lacZ* cell line overexpression (forward vs. reverse constructs). (A) Sample sizes and numbers of DEGs discovered by DESeq2, edgeR, or limma as well as common for all three with cutoff as adjusted P-Value ≤ 0.05. Numbers of DEGs discovered by DESeq2 with cutoffs as adjusted P-Value ≤ 0.05 and fold change ≥ 1.1, 1.2, 1.3, 1.4, 1.5, or 2 are also listed. Note that the only DEG always discovered in all *A930004D18Rik* datasets is *A930004D18Rik* itself; and *A830005F24Rik* was not discovered as a DEG in all *A830005F24Rik* datasets because it has very low expression in some of them and was filtered out before the statistical tests. (B) and (C) show curves of fold change (B) and dispersion (C) against fragment count of these RNA-Seq datasets, fitted with locally estimated scatterplot smoothing (LOESS) method. Values are taken from DESeq2 (fragment count as baseMean, fold change as $2^{|log2FoldChange|}$, and dispersion; S7 Table).

Hence, we used the 12.5-day whole embryos and heads of postnatal 0.5-day pups for RNA--Seq analysis.

RNA was obtained from 10 to 14 12.5-day embryos of the four sex (female or male) and genotype (homozygous knockout or wildtype) combinations. On average, we could map 67.1 million unique reads per sample (range from 36.9 to 92.7 million reads; S6 Table). First, we examined whether the *A930004D18Rik* transcript was indeed lacking in the knockouts. This is the case: knockouts show no transcription, but wildtypes show clear transcription. We also confirmed their genotypes by checking the level of *lacZ* expression. The mapped reads show the *lacZ* expression initiates with the promotor of Isoform2 (also for the RNA-Seq data from pup heads below). We found 6,173 DEGs between male knockout and wildtype samples (DESeq2, adjusted P-Value ≤ 0.05, fold changes range from 0.53 to 1.59 when excluding *A930004D18Rik* itself; Fig 3A and S7 Table) and 8,239 between females (DESeq2, adjusted P-Value ≤ 0.05, fold changes range from 0.53 to 1.56 when excluding *A930004D18Rik* itself; Fig 3A and S7 Table). Among them, there are 5,008 shared between male and female samples.

We also sequenced the heads of 10 postnatal 0.5-day pups from each of the four sex (female or male) and genotype (homozygous knockout or wildtype) combinations. On average, we could map 74.6 million unique reads for each sample (range from 59.3 to 89.4 million reads; S6 Table). Again, we confirmed that the *A930004D18Rik* transcript was indeed lacking in the knockouts, and checked the level of *lacZ* expression. We found 4,317 DEGs between male knockout and wildtype samples (DESeq2, adjusted P-Value ≤ 0.05, fold changes range from 0.65 to 1.36 when excluding *A930004D18Rik* itself; Fig 3A and S7 Table). Interestingly, we found only one DEG between females, *A930004D18Rik* itself (DESeq2, adjusted P-Value ≤ 0.05; Fig 3A and S7 Table). This can be ascribed to a higher dispersion in the female samples (Fig 3C), which results in a loss of power. Among the 4,317 DEGs between male head samples and the 6,173 ones between male embryo samples, 1,274 are overlapping.

### *A830005F24Rik* knockout effect on the transcriptome

*A830005F24Rik* is expressed in brain tissues at different stages (Fig 1C) and we targeted the RNA-Seq analysis to the heads of postnatal 0.5-day pups. We sequenced the heads of 10 individuals each of the four sex (female or male) and genotype combinations (homozygous knockout or wildtype). On average, we could map 64.7 million unique reads for each sample (range from 57.0 to 74.4 million reads; S6 Table). We confirmed that the *A830005F24Rik* transcript was indeed lacking in the knockouts, and checked the level of *lacZ* expression. We found 2,733 differentially expressed genes between male knockout and wildtype samples (DESeq2, adjusted P-Value ≤ 0.05, fold changes range from 0.72 to 1.38; Fig 3A and S7 Table), but only 820 between females (DESeq2, adjusted P-Value ≤ 0.05, fold changes range from 0.76 to 1.33; Fig 3A and S7 Table). We found also here a higher dispersion among the female samples compared to the corresponding male samples (Fig 3C) that is expected to lower the power of detection. We find 154 DEGs shared between male and female samples.

Given the bidirectional transcription of *A830005F24Rik* and its neighboring gene *Zfp169* (Fig 1B), we checked whether the latter is affected in the *A830005F24Rik* knockout construct. This is indeed the case. *Zfp169* is expressed lower in the *A830005F24Rik* knockout mice (0.78 fold in males and 0.76 fold in females). *ZFP169* belongs to the KRAB-Zn-finger proteins, which may influence the transcription of other genes through its binding to transposable elements to silence them [24–26]. Hence, we cannot exclude that this small expression difference could at least partially cause the effects we see in the *A830005F24Rik* knockouts (see Discussion).

## Further validation

All the differential expression analyses of our RNA-Seq datasets described above were performed using DESeq2. In order to be sure that the discovered DEGs are not method-specific, we also used two other generally employed methods: edgeR and limma. Similar numbers of DEGs were discovered and they are highly overlapping (Fig 3A and S7 Table for DESeq2, S8 Table for edgeR, and S9 Table for limma). The only larger difference is in dataset *A830005F24Rik* head female where somewhat different numbers of DEGs were discovered for the three methods (820 for DESeq2, 522 for edgeR, and 142 for limma). 139 out of 142 DEGs discovered by limma were also discovered by DESeq2 and edgeR (Fig 3A).

In addition, we also evaluated our RNA-Seq datasets using simulation studies by repeating differential expression analyses on randomly generated, size matched, and genotype evenly distributed sample sets with DESeq2. For each of the actual datasets of which we discovered more than one DEG, the majority of the numbers of DEGs in the simulated datasets is zero, and the actual numbers of DEGs are significantly higher than those in the simulations (S10 Table). This further proves that our differential expression analyses are valid.

To assess whether any possible effects on the transcriptome could be caused by the expression of *lacZ* in the knockout constructs, we conducted a control experiment in cell culture. We transformed primary mouse embryonic fibroblasts with vectors expressing transcripts containing the *lacZ* ORF in forward and reverse direction. This was done in 10 parallels for each direction and RNA-Seq data were obtained for each of them after 48 hrs incubation (*i.e.*, transient expression). The expression of the transcripts including the *lacZ* ORF in the forward and the reverse directions were confirmed by the uniquely mapped reads. On average we could map 54.2 million unique reads per sample (range from 44.2 to 65.8 million reads; S6 Table). We did not detect any significantly differentially expressed mouse genes in this experiment (S7–S9 Tables). Note that the overall dispersion of this *lacZ* dataset is the lowest, which indicates that no discovered DEG is not due to lack of power (Fig 3B and 3C). This suggests that LacZ protein expression by itself does not result in traceable changes of the transcriptome. This conclusion applies of course only to this particular experiment and it could be useful to eventually repeat this in a whole mouse background. However, another control already inherent in our data is that in the RNA-Seq data of the heads of postnatal 0.5-day male pups in the two lines. Both of these express *lacZ* but the sets of differentially expressed genes are different (they overlap only in 375 genes, whereby 385 would have been expected by chance; see Methods).

## Tracing possible functions

While both knockout lines show phenotypic effects, these provide no direct clue towards the possible function of the genes within their regulatory network. To explore this, we assessed the GO enrichment terms for each of the datasets. To avoid bias effects towards long or highly expressed genes, we used GOseq [23] that corrects for these problems. We found that each dataset generated a different list with up to hundreds of enriched terms (corrected P-Value $\leq$ 0.05; S11 Table). However, when focusing on overlapping terms between the datasets, we found restricted sets of eight terms for each of the genes (Table 3). *A930004D18Rik* yields terms that suggest an involvement in developmental processes via being part of protein complexes. *A830005F24Rik* yields terms that suggest a role in extracellular matrix formation.

In order to understand how larger sample size versus deeper sequencing would benefit tracing possible functions, we further identified and analyzed the overlapping enriched GO terms among *A930004D18Rik* samples (listed in Table 3) in the subsampling datasets

**Table 3. Overlapping enriched GO terms among *A930004D18Rik* samples and between *A830005F24Rik* samples.**

| Ontology[a] | Category | Term | Corrected P-Value (BH) | | | |
|---|---|---|---|---|---|---|
| | | | embryo male | embryo female | head male | head female |
| *A930004D18Rik* | | | | | | |
| BP | GO:0032501 | multicellular organismal process | 4.3E-10 | 9.5E-07 | 4.3E-02 | NA |
| MF | GO:0005515 | protein binding | 4.7E-07 | 2.9E-05 | 3.3E-07 | NA |
| BP | GO:0048856 | anatomical structure development | 1.6E-06 | 3.6E-06 | 5.9E-03 | NA |
| BP | GO:0032502 | developmental process | 7.3E-06 | 3.0E-06 | 2.8E-02 | NA |
| MF | GO:0005488 | binding | 3.5E-05 | 2.0E-05 | 5.7E-04 | NA |
| BP | GO:0048869 | cellular developmental process | 4.5E-03 | 2.1E-03 | 3.7E-02 | NA |
| MF | GO:0044877 | protein-containing complex binding | 1.0E-02 | 4.2E-03 | 2.9E-03 | NA |
| CC | GO:0032991 | protein-containing complex | 4.6E-02 | 3.2E-08 | 2.8E-02 | NA |
| *A830005F24Rik* | | | | | | |
| CC | GO:0031012 | extracellular matrix | NA | NA | 1.3E-13 | 1.4E-02 |
| CC | GO:0062023 | collagen-containing extracellular matrix | NA | NA | 6.6E-13 | 2.0E-02 |
| CC | GO:0044421 | extracellular region part | NA | NA | 1.1E-07 | 1.7E-03 |
| CC | GO:0005576 | extracellular region | NA | NA | 1.1E-06 | 6.0E-04 |
| CC | GO:0044420 | extracellular matrix component | NA | NA | 7.0E-04 | 3.2E-02 |
| BP | GO:0001649 | osteoblast differentiation | NA | NA | 8.7E-04 | 2.7E-02 |
| BP | GO:0022414 | reproductive process | NA | NA | 1.4E-02 | 1.2E-02 |
| BP | GO:0000003 | reproduction | NA | NA | 1.5E-02 | 1.2E-02 |

[a]MF: molecular function, BP: biological process, and CC: cellular component.

on sample size and sequencing depth described in Fig 2 and the related text (see power analysis for transcriptomic data). Interestingly, we found sample size is very crucial for tracing possible functions. The larger the sample size, the higher the probability that the eight enriched GO terms can be identified (Fig 4A). When sample size is only three (a number that is currently used in many studies), zero of the eight terms can be identified in more than one third (34.4%) of the subsampling datasets. In contrast when sample size is ten, at least one of the eight terms can be identified in all the subsampling datasets (Fig 4C). Intriguingly, sequencing depth is less relevant to the level of tracing possible functions in the range of our analyses, *i.e.*, in the majority of the subsampling datasets, eight enriched GO terms, can always be identified, even when the subsampling ratio is as low as 0.1 (Fig 4B and 4D).

We further performed transcriptional activity analyses on our datasets with DoRothEA [27,28] to get additional hints on possible functions. Note that, unlike GOseq, DoRothEA does not consider the bias effects towards long or highly expressed genes, and thus the results need to be interpreted more cautiously. We found that each dataset generated a different list with up to dozens of enriched regulons (corrected P-Value ≤ 0.05; S12 Table). However, when focusing on overlapping terms between the datasets, we found restricted sets of ten regulons for *A930004D18Rik* and 11 for *A830005F24Rik* (Table 4). Four regulons (*Sp1*, *Smad3*, *Nfkb1*, and *Ets1*) exist in both restricted sets, which can be either true signals or false positives due to the bias. On the other hand, unique regulons in the different restricted sets maybe less likely to underly a bias. According to the annotation in the NCBI Gene database ("https://www.ncbi.nlm.nih.gov/gene"), regulons uniquely yielded from *A930004D18Rik* suggest an involvement in development (*Sox2*, *Prdm14*, and *Esr1*); and regulons uniquely yielded from *A830005F24Rik* suggest a role in cancer (*Jun*, *Fos*, *Tfdp1*, *E2f1*, *E2f2*, *E2f4* and *Lef1*).

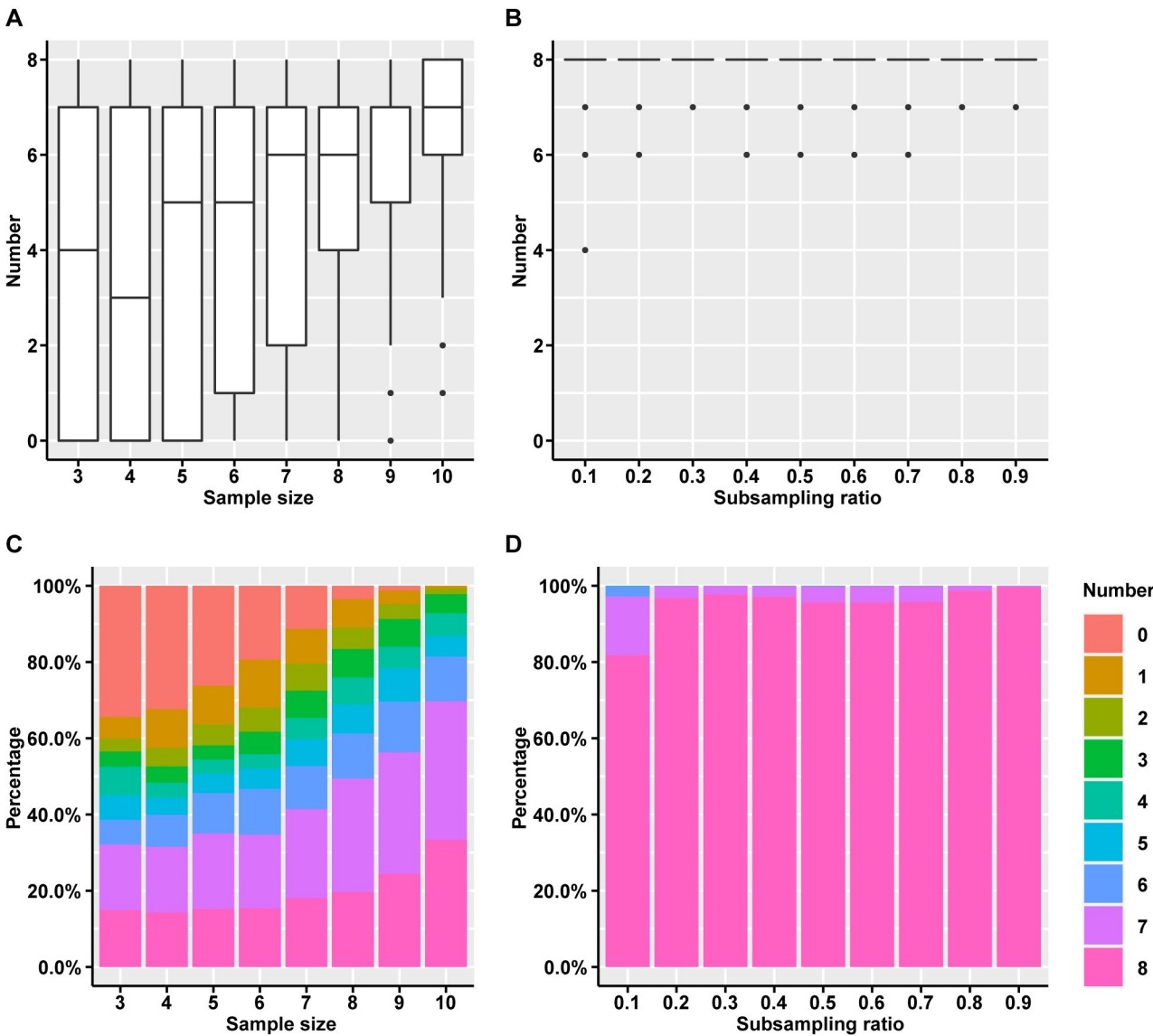

**Fig 4. Identification of the eight overlapping enriched GO terms among *A930004D18Rik* samples in the subsampling datasets.** The numbers of identified overlapping enriched GO terms among *A930004D18Rik* samples (listed in Table 3) in the subsampling datasets described in Fig 2 on sample size (A) and sequencing depth (B) are shown in box plots (data in S2–S4 Files). The percentages of the subsampling datasets on sample size (C) and sequencing depth (D) with different numbers of the identified terms are shown in bar plots.

## Discussion

We have explored here the boundary conditions for transcriptome analysis of mouse knockout strains that show no overt phenotype. We have chosen two KO strains of genes for which no hypothesis for gene function was available, *i.e.*, homology or domain searches had not yielded a match. Both genes are expressed in the brain and standardized behavior tests did reveal small behavioral effects. One gene shows an additional high expression in the developing limbs and morphological analysis of the limbs in knockouts revealed also a small effect on the phenotype. However, this phenotyping does not yield further insights into the possible molecular functions of the genes. Hence, we explored deep transcriptome sequencing of the tissues and stages at which these genes are expressed.

**Table 4. Overlapping enriched regulons among _A930004D18Rik_ samples and between _A830005F24Rik_ samples.**

| Regulon | Corrected P-Value (BH) | | | |
|---|---|---|---|---|
| | embryo male | embryo female | head male | head female |
| _A930004D18Rik_ | | | | |
| Zfp263 | 3.5E-34 | 1.5E-71 | 8.7E-45 | NA |
| Sox2 | 2.7E-13 | 1.4E-06 | 7.7E-11 | NA |
| Prdm14 | 2.7E-13 | 9.4E-31 | 6.9E-23 | NA |
| Cebpa | 4.9E-05 | 6.0E-03 | 1.0E-02 | NA |
| Esr1 | 1.0E-04 | 8.4E-06 | 1.2E-03 | NA |
| Sp1 | 1.3E-04 | 9.8E-03 | 2.8E-05 | NA |
| Smad3 | 4.7E-03 | 4.4E-05 | 2.0E-02 | NA |
| Stat3 | 6.5E-03 | 8.8E-03 | 6.3E-03 | NA |
| Nfkb1 | 1.6E-02 | 3.4E-02 | 4.0E-02 | NA |
| Ets1 | 1.8E-02 | 8.8E-03 | 2.0E-02 | NA |
| _A830005F24Rik_ | | | | |
| Sp1 | NA | NA | 2.1E-06 | 2.0E-03 |
| Jun | NA | NA | 2.1E-06 | 8.7E-04 |
| Smad3 | NA | NA | 2.7E-04 | 5.3E-05 |
| Ets1 | NA | NA | 2.0E-03 | 3.2E-03 |
| Nfkb1 | NA | NA | 4.1E-03 | 3.0E-02 |
| Fos | NA | NA | 4.1E-03 | 3.0E-02 |
| Tfdp1 | NA | NA | 8.3E-03 | 3.9E-05 |
| E2f2 | NA | NA | 1.1E-02 | 6.1E-05 |
| E2f4 | NA | NA | 1.5E-02 | 1.1E-11 |
| Lef1 | NA | NA | 1.6E-02 | 3.2E-03 |
| E2f1 | NA | NA | 1.9E-02 | 6.1E-05 |

## Setting the boundary conditions

The experimental design for comparative gene expression studies is usually more guided by practicability rather than _a priori_ power considerations. They often consist of 3–6 biological replicates and a filtering out of all genes that show less than 2-fold changes. For RNA-Seq experiments, a further consideration is the read depth and this, as well as the number of replicates, is often bounded by cost considerations. However, with the continuous fall of sequencing costs, power considerations should start to guide the design of the experiments. We have done a series of simulations that show that with still reasonable sequencing depth and number of replicates, one can achieve a much deeper resolution, obviating the need to set an artificial cutoff for the fold-change. One could of course argue that one wants to see only the strongest effects and treat the other effects as noise, but this leads to the non-satisfying situation that one has no further insights into genes that show no such strong effects. This problem has been recognized at the level of phenotyping of KO strains, where extensive phenotyping pipelines are now used to trace possible function of genes that show no overt phenotype. We argue here that this should also be extended to the transcriptome level.

Interestingly, a major result of our analysis suggests that increasing sample size always benefits the power to discover DEGs, to find enriched GO terms, and to trace possible functions, while sequencing depth tends to reach saturation in all cases, especially in tracing possible functions. This suggests that including larger numbers of biological replicates is much more important for transcriptomics studies than increasing sequencing depth.

## Transcriptome changes in weak effect loci

The most interesting direct outcome for the studied genes is that the expression of a large number of other genes is significantly affected in the knockouts when one applies experimental conditions with a high power of detection. Of course, this should not be interpreted to mean that the new genes interact directly with all of these other genes. Instead, we expect that even a single or a few interactions with other genes that are already part of a network could trigger a disturbance of the whole network. Since our experimental design allowed a very high sensitivity to detect this, we were able to see such a disturbance of many further interacting genes.

But are these small changes relevant for the function of the genes? This is a question that cannot really be answered yet. Some clues come from our increasingly better understanding of quantitative trait phenotypes. Most of these phenotypes can be much better explained by invoking a large number of small effect genes, rather than a few large effect ones. In its extreme formulation, one can even propose that the whole set of genes expressed in a given tissue contributes in some way to a given quantitative phenotype [29]. Intriguingly, this implies that modifier genes have in sum a stronger effect on the phenotype than the core network genes. Hence, this so-called omnigenic model is well suited to understand our results with respect to finding so many significant changes.

A second very interesting outcome is the seemingly major difference in effects between male and female pups. For *A930004D18Rik*, we find several thousand significantly changed genes in the males, but practically none in females (apart of the knockout gene itself). For *A830005F24Rik*, we find at least 4-fold fewer significantly changed genes in females versus males. These effects can be ascribed to the higher dispersion of the data in females, *i.e.*, the variances between individuals are much higher than in males. This variance obscures the specific effects and one would need to apply a much higher sampling size or much deeper sequencing to resolve them. While a higher dispersion in females is expected for adult females given their periodic estrous cycles, it is unexpected to find this already at the pup stage, where the hormonal cycles should not yet be active. On the other hand, the developmental sex-determination cascade is already completed at this stage, *i.e.*, different sex-specific regulatory trajectories are possible. Interestingly, although we find the higher dispersion for females versus males for both genes, the average level of dispersion is lower for *A830005F24Rik* (Fig 3C), implying that there may also be locus-specific effects causing this. It will certainly be of interest to investigate this further, but until this effect is better understood, it might be advisable to preferentially focus on changes seen in male pups only (note that the dispersion difference is much lower in embryos–Fig 3C).

## Assigning gene functions

Given that the knockout constructs were designed to generate a replacement of whole gene regions, it is difficult to infer which part of the replaced region may have conveyed the observed effects. The transcript structure of *A930004D18Rik* turns out to be much more complex than originally annotated, including two possible ORFs that may both be active, as well as the antisense long non-coding RNA. Hence, there may be overlapping effects from different genes at this locus and one could speculate that this contributes also to the higher dispersion effect seen at this locus (see above). The *A830005F24Rik* transcript is part of a bidirectional transcription unit and the replacement construct affects also to some degree the transcription of this neighboring gene, which is itself a possible transcriptional regulator. Hence, it could be the changed expression of this gene that contributes to the DEGs detected.

Given these problems, one should use different knockout designs if one would repeat these studies for the target genes. Use of CRISPR/Cas can induce short frameshifts in

ORFs that should be much more specific and reduce the risk of secondary effects. However, for the present study we consider this as a secondary problem, since even the possible secondary effects are weak, *i.e.*, the proof-of-principle of a study of weak effect loci is not compromised by not being able to pinpoint the exact molecular cause. However, it may well be that overlapping effects may have contributed to the complexity of the GO term profiles and the inference on gene function refer to the whole deleted locus, rather than a specific gene.

## Conclusions

Using transcriptomic comparisons to study the function of knockout genes appears to be a suitable extension in the repertoire of systematic phenotyping efforts for genes that show no overt knockout effects. Our power simulations and experimental verification provide the boundary conditions under which this can be done. An important conclusion from this is that increasing the number of biological replicates is superior to increasing sequencing depth. We believe that our transcriptome analyses and results, especially the power analysis, are also suitable for all transcriptome case-control studies. More complicated transcriptome experiments, such as comparisons among multiple sample groups, could also benefit from similar extended power analysis.

## Methods

### Ethics statement

The behavioral studies were approved by the supervising authority (Ministerium für Energiewende, Landwirtschaftliche Räume und Umwelt, Kiel) under the registration numbers V244-71173/2015, V244-4415/2017 and V244-47238/17. Animals were kept according to FELASA (Federation of European Laboratory Animal Science Association) guidelines, with the permit from the Veterinäramt Kreis Plön: 1401-144/PLÖ-004697. The respective animal welfare officer at the University of Kiel was informed about the sacrifice of the animals for this study.

### Mouse knockout lines

The line with allele *A930004D18Rik*$^{tm1a(EUCOMM)Wtsi}$ (genetic background: C57BL/6N) was obtained from the European Mouse Mutant Archive (EMMA). In this line the genomic DNA sequences at chr2:18,024,906–18,028,528 (mm10), including the ORFs, were substituted by a DNA fragment composed of a *lacZ* gene, the first *loxP* site, a neomycin resistance gene (*neo*), the second *loxP* site, the original genomic DNA at this locus (critical exon), the third *loxP* site (ordered from 5' to 3' on the reverse strand of mm10). We converted it to the *A930004D18Rik* knockout line (*tm1b*) using a cell-permeable Cre recombinase in order to delete the coding exon together with the *neo* gene according to the method described in [30]. In brief, the females from the line were super-ovulated and were then mated with the males from the line. The zygotes were treated with HTN-Cre from Excellgen (Catalog no. RP-7) and transferred into 0.5-day pseudo-pregnant females in the 2-cell stage. The alleles of the pups were confirmed by PCR and Sanger sequencing.

The knockout line for *A830005F24Rik* with allele *A830005F24Rik*$^{tm1.1(KOMP)Mbp}$ (genetic background: C57BL/6N) was obtained from the Knockout Mouse Project (KOMP). The genomic DNA sequences at chr13:48,514,226–48,514,747 (mm10), containing 502 of 504 nucleotides of the ORF, were substituted by a *lacZ* gene.

Primers for genotyping the two lines are listed below.

| Line | Allele (Fragment length) | Direction | Sequence (5' > 3') |
|---|---|---|---|
| *A930004D18Rik* | KO (380 bp) | Forward | CGGTCGCTACCATTACCAGT |
| | | Reverse | ACTGATGGCGAGCTCAGACC |
| | WT (323 bp) | Forward | AGAGCAAACGTGCTGGAGTG |
| | | Reverse | GCTTGGGCGATTGTGTCTC |
| *A830005F24Rik* | KO (618 bp) | Forward | GCTACCATTACCAGTTGGTCTGGTGTC |
| | | Reverse | CAAGTGCTCTTAACACTCGGTAGCC |
| | WT (331 bp) | Forward | CCTGGAAATGGTTTCATCTTGATAGG |
| | | Reverse | CAAGTGCTCTTAACACTCGGTAGCC |

## RNA-Seq and data analysis

The heads of postnatal 0.5-day pups for both lines and the 12.5-day *A930004D18Rik* embryos were carefully collected and immediately frozen in liquid nitrogen. Samples from the wildtype littermates were used for each comparison against knockouts. Total RNAs were purified using QIAGEN RNeasy Microarray Tissue Mini Kit (Catalog no. 73304). The RNA samples (including the ones from *lacZ* overexpression experiment described below) were further processed using the Illumina TruSeq Stranded mRNA HT Library Prep Kit (Catalog no. RS-122-2103), and sequenced using Illumina NextSeq 500 and NextSeq 500/550 High Output v2 Kit (150 cycles) (Catalog no. FC-404-2002). All procedures were performed in standardized and parallel reactions to minimize experimental variance.

Raw sequencing outputs were converted to FASTQ files with bcl2fastq (2.17.1.14), and reads were trimmed with Trimmomatic (0.35) [31]. Only paired-end reads left were used for following analyses. Trimmed reads were mapped to the mouse genome GRCm38 [32,33] with HISAT2 (2.0.4) [34] and SAMtools (1.3.1) [35], and taking advantage of the mouse gene annotation in Ensembl (Version 86) by using the—ss and—exon options of hisat2-build. Sample details are provided in S6 Table. We counted fragments mapped to the genes annotated by Ensembl (Version 86) with HTSeq (0.6.1p1) [36], and performed differential expression analysis with DESeq2 (1.14.1) [37]. Besides the DESeq2 default outputs, we also added the dispersions estimated by DESeq2 (1.14.1) into the outputs (S7 Table). Additionally, we also used edgeR with the exact test (3.16.4) [10] (S8 Table) and limma with the voom approach (3.30.6) [38] (S9 Table) for differential expression analysis.

When we compared two or three sets of DEGs discovered from two datasets or three methods, also required same directions of differences for shared DEGs, *i.e.*, the fold changes (KO/WT) of a shared DEG are always either larger than one or smaller than one. The expected number of shared DEGs by chance between datasets "*A930004D18Rik* head male" and "*A830005F24Rik* head male" was estimated as follows. With DESeq2 (1.14.1), 15,092 genes have enough fragments for statistical tests for both datasets. Among them, 2,193 down-regulated and 2074 up-regulated DEGs were discovered in the dataset "*A930004D18Rik* head male" (the sum of these two values, 4,267, is smaller than 4,317 in Fig 3A because 50 genes were filtered out in dataset "*A830005F24Rik* head male" due to low fragment counts); and 1,244 down-regulated and 1,489 up-regulated DEGs were discovered in the dataset "*A830005F24Rik* head male". Then the expected number of shared DEGs by chance is $((2{,}193 / 15{,}092) \times (1{,}244 / 15{,}092) + (2{,}074 / 15{,}092) \times (1{,}489 / 15{,}092)) \times 15{,}092 = 385$.

## Downstream functional analysis of RNA-Seq data

GOseq (1.38.0) [23] was used for GO enrichment analysis because the fragment count bias of the power for detecting DEGs was taken into account in this method. The outputs can be

found in S11 Table. DoRothEA (version 2) [27,28] was used for transcriptional activity analysis together with biomaRt (2.42.0) and viper (1.20.0). The TF regulon gene set used in the analysis was downloaded from "https://github.com/saezlab/ConservedFootprints.git" on 31.03.2020. According to the performance analysis in [28], the specific gene set, "data/dorothea_bench-mark/regulons/regulons_in_viper_format/dorothea_mouse_AB_viper_format.rds", was used because it produces the best performance. The outputs can be found in S12 Table.

## Power analysis for RNA-Seq data

RNASeqSampleSize (1.6.0) [11] was used for conducting the power analysis. Specifically, we used the "est_power" function, and set parameters w (ratio of normalization factors between two groups) as 1, alpha (significance level) as $3.3 \times 10^{-6}$. Then we traversed all 12,144 possible combinations of n (sample size) from 3 to 13, rho (fold change) from 1.05 to 1.5, lambda0 (read count) from 64 to 65,536, and phi0 (dispersion) from 0.0005 to 0.016 to calculate the power values (S4 Table).

## Random subsampling studies for RNA-Seq data

We performed two sets of random subsampling studies on our RNA-Seq dataset "*A930004D18Rik* embryo female". The first set of subsampling is on sample size. We subsampled N knockout and N wildtype samples (N = 3, 4, 5, 6, 7, 8, 9, 10), and identified DEGs with DESeq2 (1.14.1) [37], and performed GO enrichment analysis with GOseq (1.38.0) [23] each time, and repeated 1,000 times for each N value. The second is on sequencing depth. We subsampled fragments (a fragment is a pair of reads) from total fragments of all genes in each sample with a ratio (ratio = 0.1, 0.2, 0.3, 0.4, 0.5, 0.6, 0.7, 0.8, 0.9), and identified DEGs with DESeq2 (1.14.1) [37], and performed GO enrichment analysis with GOseq (1.38.0) [23] each time, and repeated 1,000 times for each ratio value. Numbers of DEGs are provided in S5 Table. GOseq outputs are provided in S2–S4 Files.

## *lacZ* overexpression

Primary mouse embryonic fibroblasts (MEFs) used for overexpression were obtained from C57BL/6 mice. In brief, we dissected 13.5–14.5 dpc embryos from uteruses and extraembryonic membranes into PBS (Lonza, Catalog no. BE17-512F); discarded heads and soft tissues and washed the carcasses with PBS; cut the carcasses into 2–3 mm pieces; transferred them into 50 ml Falcon tubes and added 5–20 ml Trypsin-EDTA (Gibco, Catalog no. 25300–054); vortexed and incubated for 10 minutes at 37˚C; vortexed again and incubated for 10 minutes at 37˚C; inactivated trypsin by adding 2 volumes of medium (500 ml DMEM (Lonza, Catalog no. BE12-733F), 55 ml FBS (PAN, Catalog no. P30-3702), 5.5 ml glutamine (Lonza, Catalog no. BE17-605E), 5.5 mL penicillin (5,000 U/mL) / streptomycin (5,000 μg/mL) (Lonza, Catalog no. DE17-603)); pipetted up and down to get single cell suspension; plated cells and incubated overnight.

We separately cloned the fragment of the *lacZ* ORF from the *A830005F24Rik* knockout allele (both knockout lines have the identical *lacZ* ORF) and its reverse complement fragment into pVITRO2-neo-GFP/LacZ expression vector (Catalog no. pvitro2-ngfplacz) to replace its own *lacZ* ORF using the homologous recombination method, and then purified the plasmids with QIAGEN EndoFree Plasmid Maxi Kit (Catalog no. 12362). The replacements in the vectors were confirmed by PCR and Sanger sequencing. Ten independent transfections for each of the two plasmids into the P2 MEFs were performed separately with Amaxa Mouse/Rat Hepatocyte Nucleofector™ Kit (Catalog no. VPL-1004) according to manufacturer's recommendation. Transfected cells were grown in the medium (see above). Cells were incubated at 37˚C in 5% $CO_2$ atmosphere as a pH regulator. The protein expression of *lacZ* in *lacZ*

overexpressed cells but not in reverse *lacZ* overexpressed cells was confirmed using a β-Galactosidase Staining Kit (Catalog no. K802-250). Total RNAs from the transfected cells were purified using QIAGEN RNeasy Mini Kit (Catalog no. 74106) 48 hours after transfection.

## Simulation analyses for RNA-Seq

For each of the RNA-Seq datasets of which we discovered more than one differentially expressed gene, we performed a simulation analysis. For each of the three "head" datasets, there are 10 knockout and 10 wildtype samples. We randomly assigned five knockout and five wildtype samples as group 1 and the rest samples as group 2, and then performed differential expression analysis with DESeq2 (1.14.1) [37] between the two groups in the same procedure as the actual datasets. For the two "embryo" datasets, there are 12 knockout and 10 wildtype samples, and 12 knockout and 14 wildtype samples. We randomly assigned six knockout and five wildtype samples, or six knockout and seven wildtype samples separately as group 1 and the rest samples as group 2 before performing differential expression analysis. The numbers of DEGs in these two sets of simulations are slightly overestimated because 11 vs. 11 has larger power than 12 vs. 10, and 13 vs. 13 has larger power than 12 vs. 14. For each dataset, we repeated the simulation 1,000 times, and then counted the numbers of DEGs (adjusted P-Value ≤ 0.05), and calculated the median of the 1,000 numbers and the P-Value of the actual number based on the distribution of the 1,000 numbers. Detail is provided in S10 Table.

## PacBio Iso-Seq and data analysis

This data was generated for an ongoing project in our lab. In brief, whole brains were collected from 10 weeks old *Mus musculus* males, immediately frozen in liquid nitrogen, and sent to Max Planck-Genome-centre Cologne for RNA extraction, library preparation, and sequencing on PacBio Sequel. Importantly, TeloPrime Full-Length cDNA Amplification Kit (Lexogen, Austria) was used for generating full length cDNA, with the selection of both Poly(A) tail and 5' Cap [39]. Raw data was converted to error corrected reads of CCSs with PacBio pbccs (6.0) analysis package. The CCSs were then processed according to the Iso-Seq3 pipeline (https://github.com/PacificBiosciences/IsoSeq) without performing the "clustering" step. The output full-length non-chimeric (FLNC) strand-specific reads were mapped with Blat (36) [40] using a two-step strategy in order to saving computing resource: all reads were first mapped to the two locus of *A930004D18Rik* and *A830005F24Rik*; and then the mapped reads were mapped again to the mouse reference genome GRCm38 [32,33].

## Genomic sequences for ORF determination

The genomic sequences of *Mus musculus* (GRCm38.p6), *Mus spretus* (SPRET_EiJ_v1), *Mus caroli* (CAROLI_EIJ_v1.1), *Mus pahari* (PAHARI_EIJ_v1.1), and *Rattus norvegicus* (Rnor_6.0) were retrieved from Ensembl (Version 98) [33]. For *A930004D18Rik* ORF1, the genomic sequences of *Mus spicilegus*, *Mus mattheyi*, and *Apodemus uralensis* were retrieved from the whole genome sequencing data in [41], and this locus is well covered by the uniquely mapped reads in the three species. For *A930004D18Rik* ORF2 and *A830005F24Rik* ORF, the genomic sequences of *Mus spicilegus*, *Mus mattheyi*, and *Apodemus uralensis* were determined by Sanger sequencing of the PCR fragments from the genomic DNAs purified with salt precipitation. The PCR primers listed below were designed according to the whole genome sequencing data of the three species in [41].

| ORF | Fragment | Species | Direction | Sequence (5' > 3') |
|---|---|---|---|---|
| *A930004D18Rik* ORF2 | 1 | *M. spicilegus* *M. mattheyi* *A. uralensis* | Forward | CGGATTAGTGGGCAAGCTCC |
| | | | Reverse | AAGCGAAACGGGCCTGAC |
| *A830005F24Rik* ORF | 1 | *M. spicilegus* *M. mattheyi* *A. uralensis* | Forward | CACTTCTTGGTTGTAACAGAAAGAC |
| | | | Reverse | GTAAACAATTTGATCTTTTCTAGGCTTAG |
| | 2 | *M. spicilegus* *M. mattheyi* *A. uralensis* | Forward | AGAAGTCAACAGGGACCAGATTC |
| | | *M. spicilegus* *M. mattheyi* | Reverse | AGAGGGCATCTGATCCTTGG |
| | | *A. uralensis* | Reverse | AGAGAGCATCTGATCCTTAGAAC |

## Behavioral tests

The following three behavioral tests were performed: elevated plus maze test, open field test and novel object test. All tests were recorded on video using a VK-13165 Eneo camera mounted directly above the experimental set-up and behaviors were measured using Video-Mot2 (TSE Systems). All tests were filmed in the same room under similar lighting conditions (less than 200 lux). All lights faced the ceiling in order to avoid any glare or reflections within the test arenas.

For the elevated plus maze we used an arena that was designed for testing wild mice. It was constructed as two perpendicular arms using PVC plastic and acrylic glass, and was 80 cm above ground. The dark arms of the maze were made with grey PVC plastic sides, with a white PVC plastic bottom. The dark arms were 50 cm long, 10 cm wide and 40 cm deep. Open arms had same dimensions, except that the walls were made of acrylic glass instead of grey plastic. For testing, each mouse was placed at the center of the arena at the beginning of the test using a transparent plastic transfer pipe. Mice were filmed inside the test arena for 5 minutes [42]. VideoMot2 (TSE Systems) was used to measure the time which the mouse spent in the dark arm, the light arm, and the center of the maze. After each experiment, the test arena was cleaned with 30% ethanol.

The open field arena was made of white PVC plastic and measured 60 x 60 cm, and the walls were 60 cm high. The arena was placed directly beneath a security camera and measurements were taken using VideoMot2 (TSE Systems). At the beginning of the experiment, the mouse was placed at the center of the arena using a transparent plastic transfer pipe. Each mouse was filmed for 5 minutes. Measurements taken during the open field test included the amount of time spent at the wall of the arena (up to 8 cm away from the wall) and the distance travel during the experiment [43]. After each experiment, the test arena was cleaned with 30% ethanol.

The novel object test was carried out in the same arena as the open field test. The arena was placed directly beneath a security camera and measurements were taken using VideoMot2 (TSE Systems). At the beginning of the experiment, the mouse was placed at the center of the arena using a transparent plastic transfer pipe along with a toy made of colored building blocks (Lego). Each mouse was filmed for 5 minutes. Measurements taken during the novel object test included the latency to investigate the novel object, the number of visits to the novel object, and the distance travel during the experiment. The number of visits to the novel object was accessed based on visits to an area of 7.5 cm around the novel object [43]. After each experiment, the test arena and novel object were cleaned with 30% ethanol.

All tested animals were adult males, age matched between knockouts and wildtypes. The genotypes were masked to the experimenters. Their ages were from 11 to 17 weeks old for *A930004D18Rik* and from 15 to 25 weeks old for *A830005F24Rik*. Each mouse with a 12-h light/dark cycle stayed in individual cage in a room with only male mice at least two weeks before measurements. All the tests were performed between 8:00 and 12:00. 40 *A930004D18Rik* mice were tested in the elevated plus maze test, open field test, and novel object test, divided into two groups (20 in Group A and 20 in Group B) and were tested on two different days for the same test. 36 *A830005F24Rik* mice were tested in the elevated plus maze test, divided into three groups (12 in Group A, 8 in Group B, and 16 in Group C) and were tested on three different days. The order of the mice to be measured in each group was randomly shuffled. All behavior scores are provided in S2 Table. Nested ranks test [44] was used for the statistical analyses to compare the parameters in each behavioral tests between knockouts and wildtypes. It is a non-parametric linear mixed model test, and uses the geno-type as the fixed effect and the group membership as the random effect. When there is just one group, it is essentially identical to one-tailed Wilcoxon rank sum test.

### Limb morphology

Mouse limbs were scanned using a computer tomograph (micro-CT-vivaCT 40, Scanco, Bruettisellen, Switzerland; energy: 70 kVp, intensity: 114 μA, voxelsize: 38 μm). Further, three-dimensional cross-sections were generated with a resolution of one cross-section per 0.038 mm. Two 3D landmarks were located at the endpoints of each limb bone using the TINA land-marking tool [45], and the linear distance between the two landmarks were calculated for sta-tistical analyses. Detailed description of landmarks for each bone was previously reported in [19]. Measurements were obtained from the right side of three forelimb bones (humerus, ulna, and metacarpal bone) and three hindlimb bones (femur, tibia, and metatarsal bone).

40 *A930004D18Rik* adult males at the age between 13–19 weeks were euthanized and mea-sured. They were genotyped in advance, age matched between knockouts and wildtypes, and then the genotypes were masked to the experimenters. The order of the mice to be measured in each group was randomly shuffled. All limb scores are provided in S2 Table. A PCA analysis on the variables of the six bone lengths was performed on the data with R function "prcomp" (parameters: center = T, scale. = T).

### Supporting information

**S1 Fig. Alignments of the ORF regions of *A930004D18Rik* and *A830005F24Rik*.**
(PDF)

**S2 Fig. Results of PCA analysis with limb lengths.**
(PDF)

**S1 Table. Genomic locations and FPKM values of *A930004D18Rik* and *A830005F24Rik*.** It contains two sheets.
(XLSX)

**S2 Table. Phenotyping scores of *A930004D18Rik* and *A830005F24Rik* lines.** It contains three sheets.
(XLSX)

**S3 Table. Phenotyping data of *A830005F24Rik* line from IMPC and preliminary tests.** It contains four sheets.
(XLSX)

**S4 Table. Power analysis data.**
(XLSX)

**S5 Table. The number of differentially expressed genes detected in the subsampling datasets of sample size and sequencing depth.** It contains two sheets.
(XLSX)

**S6 Table. Sample details of RNA-Seq.** It contains four sheets.
(XLSX)

**S7 Table. DESeq2 outputs.** It contains seven sheets.
(XLSX)

**S8 Table. edgeR outputs.** It contains seven sheets.
(XLSX)

**S9 Table. limma outputs.** It contains seven sheets.
(XLSX)

**S10 Table. Simulation data of RNA-Seq.** It contains two sheets.
(XLSX)

**S11 Table. GOseq outputs.** It contains five sheets.
(XLSX)

**S12 Table. DoRothEA outputs.** It contains five sheets.
(XLSX)

**S1 File. Supplemental Methods.**
(DOCX)

**S2 File. GOseq outputs of sample size subsampling.**
(GZ)

**S3 File. GOseq outputs of sequencing depth subsampling with ratio from 0.1 to 0.5.**
(GZ)

**S4 File. GOseq outputs of sequencing depth subsampling with ratio from 0.6 to 0.9.**
(GZ)

## Acknowledgments

The authors are grateful to J. Ruiz-Orera for generating the bam files from Ribo-Seq datasets; C. Pfeifle, A. Vock, A. Jonas, C. Medina, and S. Holz for keeping the mice used in this project; S. von Merten, C. Pfeifle, H. Harre, and W. Rasmus for helping mouse phenotyping experiments; B. Kleinhenz for helping cell culture experiments; E. Blohm-Sievers for helping mouse genotyping and Sanger sequencing; C. Burghardt, E. McConnell, and H. Buhtz for help with Illumina sequencing. The mouse line with *A930004D18Rik* targeted allele used for this project was obtained from EMMA; and the *A830005F24Rik* knockout mouse line used for this project was obtained from KOMP.

## Author Contributions

**Conceptualization:** Chen Xie, Diethard Tautz.

**Data curation:** Chen Xie.

**Formal analysis:** Chen Xie, Cemalettin Bekpen, Maryam Keshavarz, Rebecca Krebs-Wheaton, Neva Skrabar, Kristian K. Ullrich, Wenyu Zhang, Diethard Tautz.

**Funding acquisition:** Diethard Tautz.

**Investigation:** Chen Xie, Cemalettin Bekpen, Sven Künzel, Maryam Keshavarz, Rebecca Krebs-Wheaton, Neva Skrabar, Kristian K. Ullrich, Wenyu Zhang.

**Methodology:** Chen Xie, Cemalettin Bekpen, Sven Künzel, Maryam Keshavarz, Rebecca Krebs-Wheaton, Neva Skrabar, Kristian K. Ullrich, Wenyu Zhang.

**Supervision:** Diethard Tautz.

**Validation:** Chen Xie.

**Visualization:** Chen Xie.

**Writing – original draft:** Chen Xie.

**Writing – review & editing:** Chen Xie, Diethard Tautz.

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
