## [Decision Letter · Decision Letter 0]

20 Mar 2020

Dear Dr. Xie,

Thank you very much for submitting your manuscript "Taking mouse knockout strains to the transcriptomic opera" for consideration at PLOS Computational Biology.

As with all papers reviewed by the journal, your manuscript was reviewed by members of the editorial board and by several independent reviewers. In light of the reviews (below this email), we would like to invite the resubmission of a significantly-revised version that takes into account the reviewers' comments.

We cannot make any decision about publication until we have seen the revised manuscript and your response to the reviewers' comments. Your revised manuscript is also likely to be sent to reviewers for further evaluation.

Sincerely,

William Stafford Noble

Deputy Editor

PLOS Computational Biology

William Noble

Deputy Editor

PLOS Computational Biology

Reviewer's Responses to Questions

**Comments to the Authors:**

Reviewer #1: This study assesses the potential for transcriptomic analysis of mouse knockout strains with no physiological phenotype. The study is straightforwardly designed and the paper is well organized. The authors provide a systematic analysis of differential expression analysis, assessing the power to detect differentially expressed genes for multiple sample sizes. Importantly, multiple methods are used and results are clear and consistent. The authors also describe their choice in knockout genes, using an unbiased approach to potentially discover the functions of two uncharacterized genes.

The main outcomes of the paper are a demonstration that knockout strains may exhibit transcriptomic differences, even in cases without “overt” phenotypes, and a power analysis of expected effects. The first outcome strongly depends on what a researcher considers an “overt” phenotype, which are often extreme outcomes obtained from screening (e.g. lethality). That such knockouts nevertheless exhibit transcriptional effects is not surprising, but the power analysis presented in this context is informative. However, the authors do not discuss any of the existing literature on power analysis in transcriptomic studies, even while using a standard software tool to address this problem in their analysis, so it is unclear how special the mouse knockout case is in terms of power analysis.

Specific comments:

1. The title is not informative and borderline nonsensical. The introductory quote is only marginally helpful, and the metaphor remains opaque. A more scientifically precise title would be helpful.

2. The manuscript could benefit from stronger conclusions and recommendations, especially regarding power analysis of knockout mouse strains. I believe this is the most important topic in the paper.

3. The abstract refers to “one gene” and “the other”, which is a bit confusing. Although the genes do not have the most conversant names, it would be helpful for reader comprehension and future researchers if the genes were named in the Abstract.

4. I question that it is appropriate to call transcriptomic studies of knockouts a “new approach” (Page 3) after almost 20 years of such studies.

5. The paper consistently refers to “weak” phenotypes without explicitly defining what is meant by “weak”. Gene expression itself is a phenotype, which is presumably “strong” when thousands of genes are changing expression. I understand that this language reflects how many researchers tend to discuss their knockout studies, but this paper should precisely define what is meant.

6. How generalizable are these results beyond gene knockouts? While knockouts are a classic experiment to determine gene function, much current modeling work is focused on the effects of variants (both coding and noncoding) that do not entirely ablate a gene’s transcript. This is particularly important when assessing the function of disease variants from human GWAS of complex diseases, where an overt disease phenotype is not expected but transcriptomic changes may inform on modifications of disease-relevant processes.

7. P6: "We guess that IMPC conducted this knockout strategy..." Were the IMPC contacted in any attempt to validate this guess?

8. P 12: Bonferroni correction is possibly too conservative for transcriptome data, as genes are not independently expressed in tissues.

9. The origin of control mice was not clear. I assume wild-type littermates were used for comparisons, but this needs to be stated. Details on control mice, sample collection, and processing are essential to understand the power analysis and determine any potential batch effects.

Reviewer #2: The authors in this paper aim to characterise phenotypes of genes that have non-lethal phenotypes. To this end, they perform power analysis to identify the conditions, in which small levels of transcript changes can be traced. This is then applied to identify the function of two genes that do not yet have defined phenotypic effect upon knockout in mice. The authors carry out a multitude of behavioural phenotyping and observe modest effects. They further carry out transcriptional analysis to identify the changes in transcriptional networks upon knockout of the two candidate genes.

In the absence of overt phenotypes, the focus of the paper shifts to characterise the function of the two genes using transcriptomics approach. However, the authors do very little to elucidate the biological differences caused by gene knockouts using the transcriptomic data and leave much to be desired. The majority of the paper focuses on identification of optimal parameters in RNAseq analysis. The authors argue that for genes that have non-overt phenotypes more samples and deeper sequencing analysis is required to identify the underlying phenotype. To identify the optimal parameters the authors focus mainly on the number of DEGs. This is not an important criteria to be focusing on. The important criteria would be the underlying biological pathways the DEGs represent. At the least, the authors should perform enrichment analysis for each subsampling rather than reporting the number of DEGs. If the ultimate aim is to characterise the phenotype, additional analysis such as transcriptional activity analysis (using already available packages such as DoRothEA (Garcia-Alonso et al, 2019, Genome Research) can be performed. Network- propagation based approaches (again numerous approaches are available for transcriptomic datasets for this purpose) can further be used to identify the cellular pathways that are altered in knockout vs the control.

In addition, the authors explore the differentially expressed genes with fold change of 1.5, while mentioning that this space is rarely explored- this is not necessarily true. Standard practice in RNA-seq analysis is to use a combination of p-value and fold change cut off. While it is common for cut off of adjusted p-values to be 0.05, cut-offs on fold change values in RNA-seq analysis very much depend on the type of downstream analysis. Often studies will use less stringent log-fold changes and perform enrichment analysis to identify biological pathways.

I am recommending the paper be rejected at this point while suggesting to the authors to re-perform the analysis by focusing on if there biological pathways that are robustly identified regardless of sample size (e.g n=3 vs n=10) or sequencing depth.

**Have all data underlying the figures and results presented in the manuscript been provided?**

Reviewer #1: Yes

Reviewer #2: Yes

PLOS authors have the option to publish the peer review history of their article (what does this mean?). If published, this will include your full peer review and any attached files.

Reviewer #1: No

Reviewer #2: No
---

## [Decision Letter · Decision Letter 1]

28 Jul 2020

Dear Dr. Xie,

Thank you very much for submitting your manuscript "Taking mouse knockout strains to the transcriptomic opera: transcriptomics combined with power analysis lead to functional understanding of genes with weak phenotypic changes in knockout lines" for consideration at PLOS Computational Biology.

As with all papers reviewed by the journal, your manuscript was reviewed by members of the editorial board and by several independent reviewers. In light of the reviews (below this email), we would like to invite the resubmission of a significantly-revised version that takes into account the reviewers' comments.

We cannot make any decision about publication until we have seen the revised manuscript and your response to the reviewers' comments. Your revised manuscript is also likely to be sent to reviewers for further evaluation.

Sincerely,

Sushmita Roy, Ph.D.

Associate Editor

PLOS Computational Biology

William Noble

Deputy Editor

PLOS Computational Biology

Reviewer's Responses to Questions

**Comments to the Authors:**

Reviewer #1: The authors have adequately addressed the substantive criticisms in my review, and I believe this will be an informative study for those planning transcriptomic studies in the mouse.

I have two minor comments:

- The title is now scientifically informative, but I continue to think the metaphor is only relevant to readers with similar cultural backgrounds. That this metaphor is repeated at conferences, likely involving the same limited in-group of investigators, is not helpful to a broad readership.

- In response to Rev 1, Item #5, and on Page 5/47 of the manuscript, the authors refer to "traditional phenotyping". I believe this needs further clarification, as referring to "tradition" does not address specific methods. Are serum biomarker assays, immunostaining, or ELISAs considered traditional methods, or do the authors primarily mean physiological traits?

Reviewer #2: I would like to thank the authors for taking into consideration the original comment and including a number of new analysis to address some of my concerns on applicability on understanding biological processes. The additional analysis include GO terms enrichment analysis and TF activity analysis. The GO-term enrichments are presented in terms of number of overlapping terms and they corroborate with the results from number of DEGs. As the authors point out- this is expected as GO enrichments are calculated based on DEGs.

What is still lacking from the paper is the relevance of the number of DEGs and GO terms in biological studies. While I agree that number is a good indicator of power and without identifying DEGs one is not able to perform downstream analysis but is getting more number of DEGs necessarily more helpful? What I meant in my original comment and still maintain is that usually researchers are interested in specific differences in biological processes or signalling pathways that are different between control and condition. Getting a large list of enriched terms or gene lists is not always useful for identifying function and the authors themselves have pointed this out in the ‘Tracking possible function” section. The authors shortlist their large list of GO enrichment by focusing on overlapping terms between the datasets and identifying 8 enriched terms. Could the authors tie this section clearly with their power analysis section? Would the eight enriched terms have not been identified, had the analysis have not been performed with a particular setup?

**Have all data underlying the figures and results presented in the manuscript been provided?**

Reviewer #1: Yes

Reviewer #2: Yes

PLOS authors have the option to publish the peer review history of their article (what does this mean?). If published, this will include your full peer review and any attached files.

Reviewer #1: No

Reviewer #2: No
---

## [Decision Letter · Decision Letter 2]

20 Sep 2020

Dear Dr. Xie,

We are pleased to inform you that your manuscript 'Dedicated transcriptomics combined with power analysis lead to functional understanding of genes with weak phenotypic changes in knockout lines' has been provisionally accepted for publication in PLOS Computational Biology.

Best regards,

Sushmita Roy, Ph.D.

Associate Editor

PLOS Computational Biology

William Noble

Deputy Editor

PLOS Computational Biology

Reviewer's Responses to Questions

**Comments to the Authors:**

Reviewer #1: The authors have addressed all comments to my satisfaction.

Reviewer #2: The authors have performed further analysis to address some of the earlier concerns about pathways enrichments that I had. The manuscript is acceptable for publication.

**Have all data underlying the figures and results presented in the manuscript been provided?**

Reviewer #1: Yes

Reviewer #2: Yes

PLOS authors have the option to publish the peer review history of their article (what does this mean?). If published, this will include your full peer review and any attached files.

Reviewer #1: No

Reviewer #2: No

---

## [Editor Report · Acceptance letter]

29 Oct 2020

PCOMPBIOL-D-20-00099R2 

Dedicated transcriptomics combined with power analysis lead to functional understanding of genes with weak phenotypic changes in knockout lines

Dear Dr Xie,

I am pleased to inform you that your manuscript has been formally accepted for publication in PLOS Computational Biology. Your manuscript is now with our production department and you will be notified of the publication date in due course.

With kind regards,

Sarah Hammond
